# A physical model describing the interaction of nuclear transport receptors with FG nucleoporin domain assemblies

Raphael Zahn[1†], Dino Osmanović[2,3,4,5†], Severin Ehret[1], Carolina Araya Callis[1], Steffen Frey[6‡], Murray Stewart[7], Changjiang You[8], Dirk Görlich[6], Bart W Hoogenboom[2,3*], Ralf P Richter[1,9,10*]

[1]Biosurfaces Lab, CIC biomaGUNE, San Sebastian, Spain; [2]London Centre for Nanotechnology, University College London, London, United Kingdom; [3]Department of Physics and Astronomy, University College London, London, United Kingdom; [4]Department of Physics, Bar-Ilan University, Ramat Gan, Israel; [5]Institute of Nanotechnology and Advanced Materials, Bar-Ilan University, Ramat Gan, Israel; [6]Department of Cellular Logistics, Max Planck Institute for Biophysical Chemistry, Göttingen, Germany; [7]Medical Research Council Laboratory of Molecular Biology, Cambridge, United Kingdom; [8]Department of Biology, University of Osnabrück, Osnabrück, Germany; [9]Laboratory of Interdisciplinary Physics, University Grenoble Alpes - CNRS, Grenoble, France; [10]Max-Planck-Institute for Intelligent Systems, Stuttgart, Germany

**\*For correspondence:**
b.hoogenboom@ucl.ac.uk (BWH);
rrichter@cicbiomagune.es (RPR)

[†]These authors contributed equally to this work

**Present address:** [‡]NanoTag Biotechnologies, Göttingen, Germany

**Competing interests:** The authors declare that no competing interests exist.

**Abstract** The permeability barrier of nuclear pore complexes (NPCs) controls bulk nucleocytoplasmic exchange. It consists of nucleoporin domains rich in phenylalanine-glycine motifs (FG domains). As a bottom-up nanoscale model for the permeability barrier, we have used planar films produced with three different end-grafted FG domains, and quantitatively analyzed the binding of two different nuclear transport receptors (NTRs), NTF2 and Importin β, together with the concomitant film thickness changes. NTR binding caused only moderate changes in film thickness; the binding isotherms showed negative cooperativity and could all be mapped onto a single master curve. This universal NTR binding behavior – a key element for the transport selectivity of the NPC – was quantitatively reproduced by a physical model that treats FG domains as regular, flexible polymers, and NTRs as spherical colloids with a homogeneous surface, ignoring the detailed arrangement of interaction sites along FG domains and on the NTR surface.

## Introduction

In eukaryotic organisms, the nuclear envelope separates the nucleoplasm from the cytoplasm and encloses most of the genetic material in the cell. The ordered course of gene expression requires selective transport through this double membrane. This function is provided by nuclear pore complexes (NPCs), large membrane-spanning protein complexes that perforate the nuclear envelope (*Fahrenkrog and Aebi, 2003*; *Fernandez-Martinez and Rout, 2012*; *Floch et al., 2014*; *Gorlich and Kutay, 1999*; *Grossman et al., 2012*; *Macara, 2001*). Although small molecules up to 20–40 kDa (i.e., up to roughly 5 nm in diameter) can diffuse freely through the NPC, the passage of larger macromolecules is impeded unless they are bound to nuclear transport receptors (NTRs) (*Gorlich and Kutay, 1999*; *Keminer and Peters, 1999*; *Mohr et al., 2009*; *Yang and Musser, 2006*).

**eLife digest** The cells of animals, plants and other eukaryotic organisms contain a compartment called the nucleus that contains most of the cell's genetic material. Proteins and other molecules – collectively known as cargos – can enter and exit the nucleus via tiny channels in the membrane that surrounds and protects it. Receptor proteins – called nuclear transport receptors – bind to potential cargos and shuttle them through the channels.

This selective transport process relies on the nuclear transport receptors being attracted to flexible, spaghetti-like proteins that are anchored to the walls on the inside of each channel. However, because of their flexible and disordered nature, these so-called FG domains are difficult to study, and the details of the transport process are poorly understood.

Zahn, Osmanović et al. decided to study how the FG domains behave and what happens when they interact with nuclear transport receptors by using ultrathin films made of just the FG domains. This is a good model system because the films are easier to study than the whole channels, but are likely to retain the essential properties of the real barrier formed in the nuclear envelope. Zahn, Osmanović et al. compared the binding of two nuclear transport receptors of different sizes, taken from humans and yeast, to FG domain films made from one of three different FG domains. The experiments showed that the different nuclear transport receptors bind to the different FG domains in very similar ways.

Zahn, Osmanović et al. then used a computational model that essentially represented the FG domains as sticky spaghetti and the nuclear transport receptors as perfectly round meatballs. This sticky-spaghetti-with-meatballs model reproduced the experimental data, implying that the exact chemical make-up and structure of the molecules may not be critical for controlling the transport of cargo across the nuclear envelope. Future studies will test whether the generic physical features of nuclear transport receptors and FG domains can indeed explain how the cargo molecules pass through the nuclear envelope.

The permeability barrier in the 30 to 50 nm diameter central transport channel of NPCs is formed by domains of NPC proteins (nucleoporins) that are rich in phenylalanine-glycine motifs (FG domains; single-letter code is used throughout) and that are grafted to the channel walls at a high density (*Bui et al., 2013*). The FG domains are thought to be highly flexible and behave like natively-unfolded proteins (*Denning et al., 2003*; *Denning and Rexach, 2007*; *Denning et al., 2002*; *Hough et al., 2015*). As such, they do not have a defined secondary or higher-order protein structure. However, depending on their intramolecular and intermolecular interactions, these proteins can organize into supramolecular assemblies such as protein meshworks, brushes or scaffolds (*Dyson and Wright, 2005*; *Uversky and Dunker, 2010*). There is a broad consensus that FG domains interact with NTRs and facilitate their passage through NPCs. In addition, there is an attractive (cohesive) interaction between FG domains. This promotes the formation and determines the properties of FG domain phases (*Eisele et al., 2013*; *Frey and Gorlich, 2007*; *Frey et al., 2006*; *Patel et al., 2007*; *Schmidt and Gorlich, 2015*), and is also essential for the formation of a functional permeability barrier (*Frey et al., 2006*; *Hulsmann et al., 2012*). However, the nature of these interactions remains controversial (*Peters, 2009*), both for the interactions between FG domains, and for the interactions between FG domains and NTRs. As a consequence, there remains a large uncertainty about the morphology of the permeability barrier (*Frey and Gorlich, 2007*; *Frey et al., 2006*; *Lim et al., 2007*; *Peters, 2005*; *Rout et al., 2000*; *Yamada et al., 2010*) and about how it is influenced by the substantial enrichment of NTRs in the NPC conduit (*Frey and Gorlich, 2009*; *Kapinos et al., 2014*; *Lowe et al., 2015*; *Mohr et al., 2009*).

Because of the low degree of order in the FG domain meshwork and its spatial confinement within the NPC, it has been difficult to address these questions using traditional biochemical assays and structure determination methods. Alternatively, computational models can provide valuable insights into collective behavior of FG domains, but are affected by the size and complexity of the NPC, and in particular by the experimental uncertainty on protein distributions and interactions (see *Osmanovic et al., 2013a* for a review).

To obtain a more comprehensive understanding of the interactions between FG domains and NTRs in the context of nucleo-cytoplasmic transport, we have employed a novel approach that combines measurements of the uptake of NTRs by well-defined nanoscale assemblies of FG domains with (coarse-grained) computational modeling for a quantitative interpretation of the experimental results in terms of FG domain morphology and intermolecular interactions. Specifically, we produce films of end-grafted purified FG domains that are similar to the protein meshwork in NPCs, both in their thickness and in their FG motif density (*Eisele et al., 2010*; *2013*). With such planar model systems it is possible to quantify NTR binding and investigate NTR-induced thickness changes (*Eisele et al., 2010*; *2012*; *Schoch et al., 2012*). Binding curves of several NTR/FG domain systems have been shown to deviate from an ideal Langmuir isotherm, suggesting that the binding avidity of NTRs to FG domain films strongly depends on the concentration of NTRs in the film (*Eisele et al., 2010*; *Kapinos et al., 2014*; *Schleicher et al., 2014*; *Wagner et al., 2015*) and on the proportion of FG motifs that are occupied by NTRs. Here, we describe the use of this system to identify common features and obtain a more quantitative understanding of these interactions, by analyzing and comparing the binding of two different NTRs, nuclear transport factor 2 from *Homo sapiens* (NTF2) and Importin β from *Saccharomyces cerevisiae* (Impβ), to plane-grafted FG domain films that each are generated from one of three different FG domains: the FG domain of Nsp1 from *S. cerevisiae* (that has FxFG and just FG motifs), a glycosylated FG domain of Nup98 from *Xenopus tropicalis* (Nup98-glyco; with primarily GLFG and just FG motifs) and an artificial, regular repeat with exclusively FSFG motifs (reg-FSFG). The two transport receptors differ in size (29.0 kDa for the functional NTF2 homodimer and 95.2 kDa for Impβ) and in the number and distribution of binding sites for FG domains. Two identical sites are located between the subunits of NTF2 (*Bayliss et al., 2002*), whereas for mammalian Impβ two different sites have been identified by crystallography (*Bayliss et al., 2000*) and molecular dynamics simulations have suggested there may be up to nine sites spread over the Impβ surface (*Isgro and Schulten, 2005*). Recent crystallography work revealed eight binding sites on the exportin CRM1 (*Port et al., 2015*), suggesting that the dispersal of binding pockets across the protein surface is a common feature of the larger NTRs. The FG domains employed in this study differ in prevalent FG motif types, FG domain size, abundance of FG motifs relative to FG domain size (*Table 1*), as well as in the distribution of FG motifs along the peptide chains and the composition of the spacer regions between FG motifs (*Table 1—source data 1*) (*Labokha et al., 2013*; *Radu et al., 1995*; *Rout and Wente, 1994*).

Our approach has enabled us to explore the universality/diversity of NTR binding to FG domains, to quantify the binding and to interpret it in terms of NTR distribution in and on FG domain assemblies, while also demonstrating how we can benchmark parameters in computational simulations to a well-defined experimental model. From the quantitative comparison between experiment and computational modeling, we learn about the levels of structural and chemical detail and heterogeneity that are required to effectively model and understand NTR uptake by FG domain assemblies, and gain new insights into the physical mechanisms – largely related to collective low-affinity interactions

**Table 1.** Properties of employed FG domain constructs. See *Table 1—source data 1* for the full amino acid sequences of these constructs.

| FG domain | amino acids [a] | sequence | FG motifs | | | FG motifs/amino acids |
|---|---|---|---|---|---|---|
| | | | FxFG | GLFG | Other | |
| Nsp1 | 615 | irregular, natural | 19 | 0 | 14 | 0.054 |
| Nup98-glyco | 496 | irregular, natural | 3 | 8 | 28 | 0.079 |
| reg-FSFG | 315 | regular, artificial | 16 | 0 | 0 | 0.051 |

[a] Excluding the His tags but including all other auxiliary amino acids (TEV cleavage sites, Cys tags and spacers).

Source data 1. Amino acid sequence of employed FG domain constructs. FG domains are shown in black letters, His tags in blue letters, and remaining parts (i.e., TEV cleavage sites, Cys tags and spacers) in grey letters. FxFG motifs are marked in yellow, GLFG motifs in green, other FG motifs in purple. Nup98-glyco features O-GlcNAc on ~30 of the S and T residues.

and the formation of a phase (*Hyman and Simons, 2012*) of FG domains and NTRs – that determine NPC transport selectivity.

## Results

### FG domain film assembly and experimental approach

Selected FG domains, i.e., Nsp1, Nup98-glyco and reg-FSFG, were purified (*Figure 1—figure supplement 1*) and anchored stably and specifically to planar surfaces, through their His tags (*Figure 1—figure supplement 2*). We monitored the formation of FG domain films and their interaction with NTF2 and Impβ by spectroscopic ellipsometry (SE) and quartz crystal microbalance (QCM-D), simultaneously and on the same sample (*Figure 1—figure supplement 3*), to quantify areal protein densities, $\Gamma$ (i.e., amounts of protein per unit area, expressed as pmol/cm$^2$; 1 pmol/cm$^2$ equals 0.6 molecules per 100 nm$^2$), and effective film thicknesses, $d$, respectively.

The FG domain grafting density was tuned to range between 4 and 11 pmol/cm$^2$ (i.e., between 2.4 and 6.6 molecules per 100 nm$^2$), by varying the FG domain solution concentration and incubation time. This range covers and extends around the estimated grafting density in a yeast NPC that is thought to be 5.2 to 6.9 pmol/cm$^2$ (i.e., 3.1 to 4.1 molecules per 100 nm$^2$); this estimate is based on the assumption of a cylindrical channel of 35–40 nm in diameter and 30–35 nm in length (*Yang et al., 1998*), and of ~136 FG domains per channel (*Rout et al., 2000*; *Strawn et al., 2004*). It is also a range (≥5 pmol/cm$^2$ for Nup98-glyco) over which the FG domain films maintain a homogeneous appearance, as previously verified by atomic force microscopy (*Eisele et al., 2013*).

NTRs were titrated into FG domain films over a range covering three orders of magnitude in NTR concentration. This range includes the typical cellular concentrations of NTRs, e.g., 0.5 μM NTF2 homodimer in *X. laevis* eggs (*Kirli et al., 2015*), 0.3 μM NTF2 homodimer in HeLa cells (*Gorlich et al., 2003*), and 3 to 5 μM Impβ in *X. laevis* (*Kirli et al., 2015*; *Wuhr et al., 2014*). The highest concentration in our experiments (10 μM) is comparable to the total concentration of NTRs found in cells (*Hahn and Schlenstedt, 2011*; *Kirli et al., 2015*; *Wuhr et al., 2014*).

*Figure 1* summarizes the experimental data at equilibrium as a function of NTR concentration, $c_{\mathrm{NTR}}$, in solution. A set of controls confirmed that NTF2 and Impβ bound specifically to the immobilized FG domains (*Figure 1—figure supplements 4* and *5*), and that binding equilibriums were indeed achieved (*Figure 1—figure supplement 3B*). Irrespective of the FG domain and NTR types, NTR binding and unbinding was rapid, i.e., largely determined by mass transport to and from the surface upon changes in NTR concentration (*Figure 1—figure supplement 3B*). This is consistent with reports on the kinetics of NTRs interacting with individual FG motifs (*Milles et al., 2015*), FG domains (*Hough et al., 2015*), and FG domain assemblies (*Eisele et al., 2010*; *Frey and Gorlich, 2007*), which all found binding to be exceptionally rapid. Taking these observations together, we conclude that we measure genuine interactions between NTRs and supramolecular assemblies of FG domains.

### Analysis of NTR binding isotherms

Interestingly, the shape of the binding isotherms (i.e., the areal NTR density in the film, $\Gamma_{\mathrm{NTR,eq}}$, versus $c_{\mathrm{NTR}}$; *Figure 1*, *top row*) remained largely unchanged with FG domain type and grafting density. This common shape prompted a more detailed analysis, including the use of phenomenological models (see *Figure 2A* for a selected measurement; all other measurements led to similar conclusions, see *Figure 2B*). For $c_{\mathrm{NTR}} \leq 0.05$ μM, the slope in the log-log binding isotherms is one (*Figure 1*; and *Figure 2A*, main plot, *dashed line*), as expected from the low-concentration limit of a Langmuir isotherm. This indicates that – at low concentrations – individual NTR molecules bind to the FG domain film independently. In this concentration range, the ratio $\Gamma_{\mathrm{NTR,eq}} / c_{\mathrm{NTR}} = PC \times d$ was constant, with partition coefficients $PC$ between $10^3$ and $10^5$ (*Figure 2—figure supplement 1A*), implying that NTRs are strongly enriched in the FG domain films compared to their concentration in solution.

For higher concentrations, however, the Langmuir isotherm (i.e., $\Gamma_{\mathrm{NTR,eq}} = \Gamma_{\mathrm{NTR,max}} \times c_{\mathrm{NTR}} / (K_{0.5} + c_{\mathrm{NTR}})$, with $\Gamma_{\mathrm{NTR,max}}$ the maximum areal density of bound NTRs and $K_{0.5}$ the concentration for half-maximum binding) failed to faithfully describe the data (*Figure 2A*, inset, *dashed line*). This is in line with earlier observations (*Eisele et al., 2010*; *Wagner et al., 2015*). For a quantitative comparison

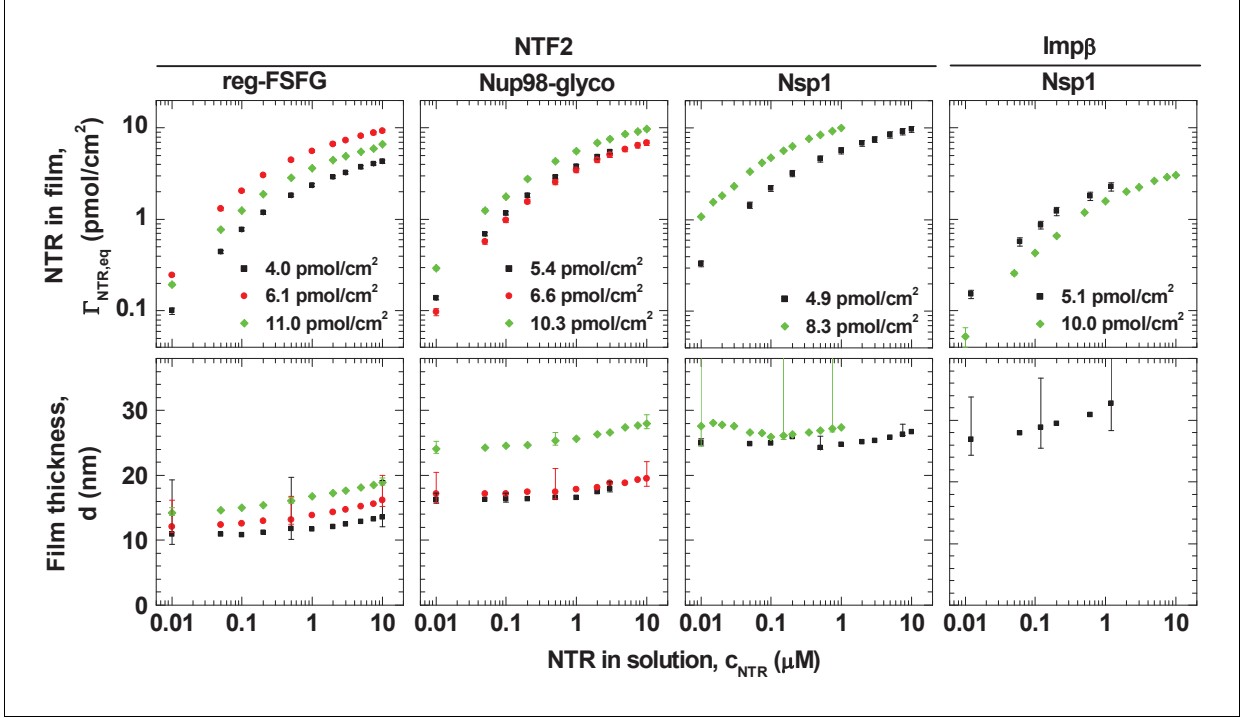

**Figure 1.** Isotherms of NTR binding (*top row*, log-log presentation) and FG domain film thickness evolution (*bottom row*, lin-log presentation) for NTF2 and Impβ binding to different FG domains (see labels at top) at selected FG domain grafting densities (visualized by distinct symbols and colors). Error bars are shown for all data points in the binding isotherms, and for three selected data points (indicating the trends) per curve in the thickness isotherms. The data for Impβ binding to the 10.0 pmol/cm² Nsp1 film were reproduced from *Eisele et al. (2010)*; this data was acquired with Nsp1 carrying a His tag at the opposite end (N-terminus) compared to the other Nsp1 data in this study, in a separate SE measurement and no simultaneously recorded thickness data are available. Full experimental details are available in 'Materials and methods' and *Figure 1—figure supplements 1–5*; tabulated results are available in *Figure 1—source data 1*.

The following source data and figure supplements are available for figure 1:

**Source data 1.** Tables of data shown in *Figure 1*.

**Figure supplement 1.** Quality of purified recombinant proteins used in this study.

**Figure supplement 2.** FG domains are anchored specifically and stably through their terminal His tag.

**Figure supplement 3.** Schematic illustration of the experimental approach and representative data.

**Figure supplement 4.** Controls for the binding of NTRs to His tag capturing surfaces monitored by QCM-D.

**Figure supplement 5.** NTF2 binds all FG domains predominantly through its primary binding site.

between different curves, we fitted the experimental data with the Hill equation (*Figure 2A*, main plot, *solid line*), i.e., $\Gamma_{NTR,eq} = \Gamma_{NTR,max} \times c_{NTR}^{\alpha} / (K_{0.5}^{\alpha} + c_{NTR}^{\alpha})$ (*Weiss, 1997*). The Hill coefficients for all curves lie within the narrow range $\alpha = 0.71 \pm 0.04$ (*Figure 2—figure supplement 1B*). This narrow spread in $\alpha$ in the Hill fits and the small variations (typically less than a factor of two) in $K_{0.5}$ for the different FG domain types and grafting densities (*Figure 2—figure supplement 1C*), confirm the uniformity of the binding isotherms noted above. Unsurprisingly, there was more variation in the effective maximal binding $\Gamma_{NTR,max}$ as determined from the Hill fits (*Figure 2—figure supplement 1D*). The uniformity of the binding isotherms can be further articulated by plotting normalized areal densities ($\Gamma_{NTR,eq}/\Gamma_{NTR,max}$) versus normalized NTR concentrations ($c_{NTR}/K_{0.5}$), with $\Gamma_{NTR,max}$ and $K_{0.5}$ determined from the Hill fit, to show that this reduces all data to a single master curve (*Figure 2B*). The

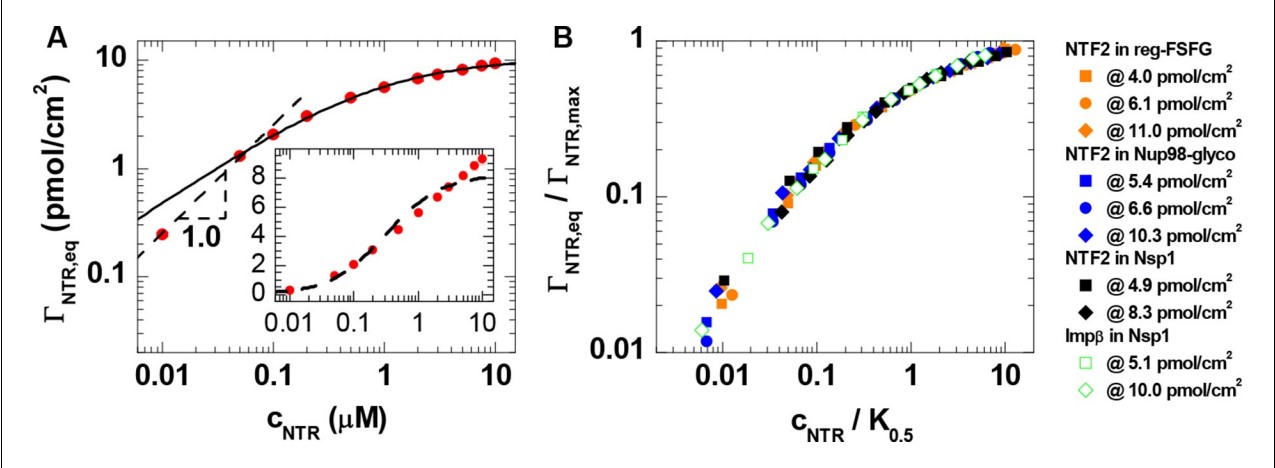

**Figure 2.** Quantitative analysis of the binding isotherms. (A) A selected data set (NTF2 binding to 6.1 pmol/cm$^2$ reg-FSFG; *symbols*) with fits to simple binding models (*lines*). Data at low NTR concentration ($c_{NTR} \leq 0.05$ μM) display a close-to-linear relation (*dashed line* with slope 1.0 in the log-log plot), as expected for independent binding, yet the Langmuir isotherm (*inset, dashed line* in lin-log plot) fails to reproduce the data over the full range of NTR concentrations. The Hill equation provides a good description of the data in the high-concentration range (0.05 μM $\leq c_{NTR} \leq$ 10 μM; *solid line*). (B) By normalizing the areal densities and NTR concentrations to $\Gamma_{NTR,max}$ and $K_{0.5}$, respectively, all data could be overlaid on a single master curve, where the effective maximal binding $\Gamma_{NTR,max}$ and the half-maximal binding $K_{0.5}$ were determined from fits with the Hill equation (see main text and *Figure 2—figure supplement 1*).

The following figure supplement is available for figure 2:

**Figure supplement 1.** Quantitative analysis of the binding isotherms in *Figure 1*.

agreement in curve shape is remarkable, given that the used FG domains provide a large spread of FG motif types and FG motif arrangement along the peptide chains (*Table 1*) and that the two tested NTR types differ both in size and in the number of binding sites for FG motifs.

Since α is smaller than one, the Hill fits indicate that NTR binding is negatively cooperative in the physiologically relevant concentration range, i.e., the average binding strength decreases as the FG domain assembly becomes enriched with NTRs. This finding is in line with recent reports that propose a modulation of NTR binding by the presence of other NTRs (*Kapinos et al., 2014*; *Schleicher et al., 2014*; *Wagner et al., 2015*). In this context, it is worth noting that the areal density of bound NTR represents only a small fraction of the FG motif density available in the films. The total number of FG motifs per FG domain is 33 for Nsp1, 39 for Nup98-glyco and 16 for reg-FSFG (*Table 1*). Our data illustrate that, at $c_{NTR} = 10$ μM, the films contained at least 10 and 50 FG motifs per bound NTF2 dimers and per bound Impβ, respectively (*Figure 2—figure supplement 1E*). With two binding sites for FG motifs per NTF2 dimer, and up to nine binding sites per Impβ, this implies that no more than 20% of the FG motifs were simultaneously engaged in NTR binding. Importantly, NTR binding does not correlate with the total abundance of FG motifs: for example, Nsp1 binds more than twice the number of NTF2 per FG motif compared to Nup98-glyco (*Figure 2—figure supplement 1E*), consistent with its binding NTF2 more strongly (*Clarkson et al., 1997*).

Taken together, the analysis of binding isotherms demonstrates that NTRs are substantially enriched in FG domain films, that the accumulation of NTRs in FG domain films progressively reduces the strength of NTR binding, and that the NTR binding behavior has universal features that are independent of the detailed chemical and structural features of the FG domains and NTRs.

## Impact of NTR binding on film thickness

Variations in film thickness $d$ (*Figure 1*, *bottom row*) following NTR binding were generally moderate. At NTF2 solution concentrations up to 1 μM, the thickness remained virtually unchanged for Nup98-glyco and reg-FSFG, and decreased marginally (by up to 7%) for Nsp1. At higher concentrations, the thickness gradually increased, by between 5 and 35% at 10 μM compared to the pristine FG domain film, depending on the FG domain type and grafting density. For Impβ binding to Nsp1,

there was a moderate and gradual thickness increase up to 25% at 10 µM. In all cases, the increase in film thickness was smaller than or comparable with the dimensions of the NTRs. These findings are in clear disagreement with the film collapse by more than 50%, reported by Lim et al. on nanoscale islands of FG domain assemblies (*Lim et al., 2007*), and the 'nanomechanical collapse' model proposed based on those data. Instead, our data are in full agreement with other thickness measurements on similar systems (*Eisele et al., 2010*; *Kapinos et al., 2014*; *Wagner et al., 2015*), which consistently did not give any indications for such a collapse, but rather indicate that the global morphology of FG domain films remains preserved irrespective of the concentration and type of NTR.

## Computational approach and cohesiveness of the FG domain films

In our experiments, the shape of the binding isotherms was independent of the detailed chemical and structural features of the FG domains and NTRs. We therefore hypothesized that it must arise from generic physical features of the FG domain / NTR system, among which are the nature of FG domains as flexible polymers and of NTRs as globular colloids, as well as the mean, overall interactions between FG domains and NTRs. To test this hypothesis, we adapted a previously developed computational model (*Osmanovic et al., 2012*; *2013b*) to planar surfaces (*Figure 3A*). The model treats polymers as beads on a chain, where the bead diameter is set to twice the contour length of an amino acid, to reproduce the flexibility of unfolded peptide chains (see Materials and methods); the interactions between FG domains are essentially smeared out over the whole (homogeneous) chain thus effectively including interactions between FG motifs, but potentially also with other parts of the FG domain chains. The model explicitly considers the confinement through grafting, the size, the flexibility, the geometrical excluded volume and the cohesiveness of FG domains, the concentration and geometrical excluded volume of NTRs, and the attraction between FG domains and NTRs. The polymer and the colloid surface are homogeneous, and two adjustable parameters regulate the interaction strengths (see Materials and methods): $\varepsilon_{pp}$ the cohesiveness (*Eisele et al., 2013*) of polymer segments, and $\varepsilon_{pc}$ the attraction between a polymer segment and a colloid. From the computed density maps (*Figure 3—figure supplement 1*) for appropriate polymers and colloids, physical parameters such as average film thickness and binding isotherms were extracted (*Figure 3—figure supplements 2–5*) and compared with the experimental data.

First, we considered the FG domain films without NTRs. *Figure 3B* displays the predicted film thickness normalized by the number of amino acids in the protein. As expected, the film thickness decreases (i.e., the film condenses) with increasing cohesiveness $\varepsilon_{pp}$. At any given $\varepsilon_{pp}$ for the polymers representing Nsp1 and Nup98-glyco, the normalized thickness is identical, as would be expected based on mean field theory for polymer brushes, irrespective of the degree of cohesiveness (*Zhulina et al., 1990*). The somewhat larger values for the roughly two-fold shorter chain representing reg-FSFG are thus likely to reflect finite-size effects. To obtain estimates for the cohesiveness parameter $\varepsilon_{pp}$ for all FG domains, we compared the computational predictions (*symbols in Figure 3B*) to the experimental thickness data (*horizontal bands in Figure 3B*). *Table 2* shows $\varepsilon_{pp}$ as a function of FG domain type, obtained via a cubic interpolation (*dashed line in Figure 3B*) between the $\varepsilon_{pp}$ values for which computational data were available. Because the interface between the film and the bulk solution is not ideally sharp, it was unclear a priori which computational density threshold (see Materials and methods) should match the effective thickness measured by QCM-D most accurately. Reassuringly, however, the results were only weakly influenced by the precise definition of film thickness that was used in the computational model, i.e., density thresholds of 1% or 10% instead of 5% typically altered the estimates for $\varepsilon_{pp}$ by less than 10%. From the comparison of computation and experiment, it was clear that $\varepsilon_{pp}$ for Nup98-glyco exceeded that of Nsp1, in good agreement with our previous study (*Eisele et al., 2013*), in which we also found Nup98-glyco to be more cohesive than Nsp1. Nup98-glyco and reg-FSFG have similar cohesiveness.

Using similar computational modeling in a NPC-mimicking pore geometry of 50 nm diameter, we had previously estimated $\varepsilon_{pp} \approx 0.05\ k_BT$ from nanomechanical studies on intact NPCs (*Bestembayeva et al., 2015*). This is close to the $\varepsilon_{pp}$ values found here, though slightly higher, probably because the presence of NTRs was not taken into account in the computational model to which the nanomechanical data were matched.

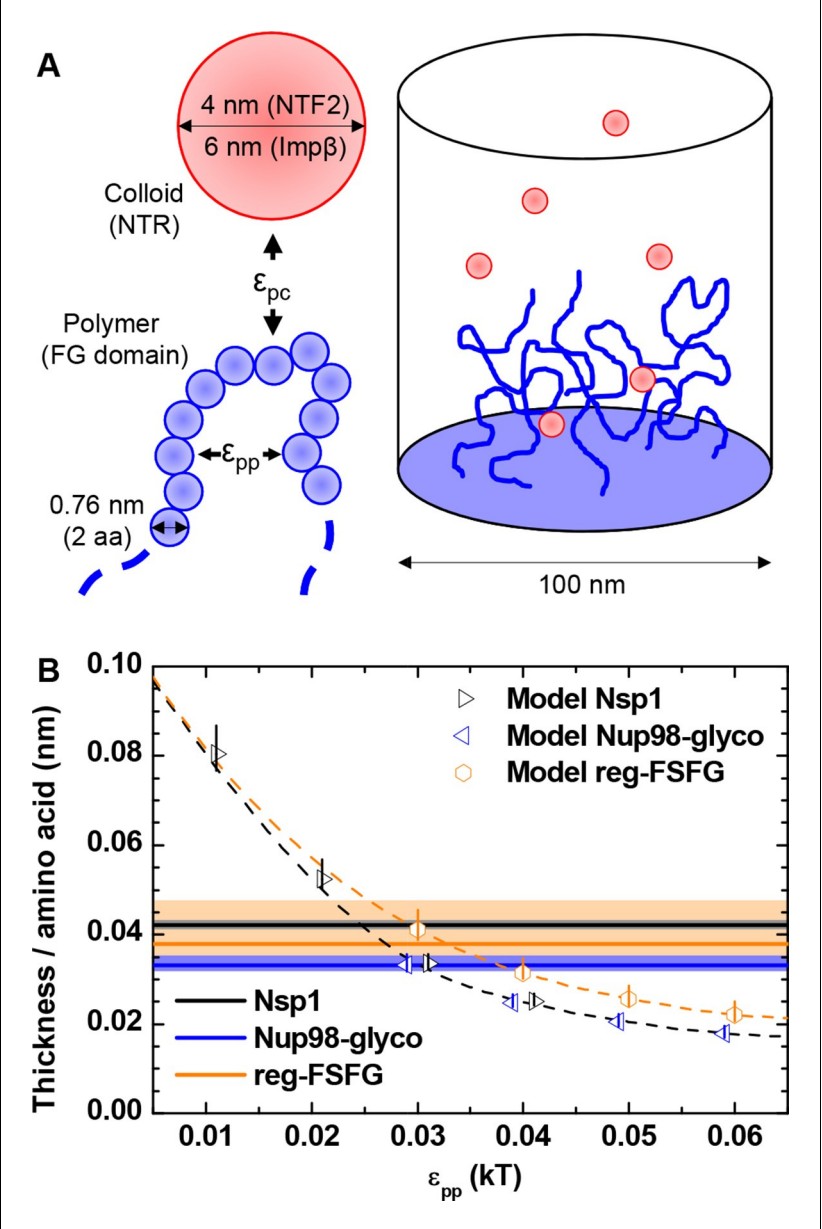

**Figure 3.** Computational model. (**A**) Schematic illustration of the computational model. FG domains are represented as end-grafted polymers anchored at 5.5 pmol/cm$^2$ (i.e., 3.3 molecules per 100 nm$^2$) to the bottom of a 100 nm diameter cylinder, and modeled as strings of beads, where each bead has equal bond length and diameter (two amino acids, 0.76 nm). The number of polymer beads was set to match the length of experimentally used FG domains. NTF2 dimers and Impβ are represented as spherical colloids of 4.0 and 6.0 nm diameter, respectively. (**B**) Matching of the computational model with experimental data for FG domain films in the absence of NTRs. Horizontal lines represent the experimentally determined film thickness per amino acid for different FG domains (*black line* - Nsp1 at 4.9 pmol/cm$^2$; *blue line* - Nup98-glyco at 5.4 pmol/cm$^2$; *orange line* - reg-FSFG at 6.1 pmol/cm$^2$), with shaded areas in matching colors indicating confidence intervals. Symbols represent the thickness as predicted by the computational model as a function of $\varepsilon_{pp}$ for the different FG domains (at 5.5 pmol/cm$^2$; colors match experimental data). The data points and the upper and lower ends of the vertical lines refer to the effective thicknesses where the densities have dropped to 5%, 1% and 10% of the maximal densities in the film, respectively. Symbols for Nsp1 and Nup98-glyco are translated along the *x* axis by +0.1 $k_B T$ and -0.1 $k_B T$, respectively, to improve their visibility. Dashed lines through the symbols are cubic interpolations (the *black dashed line* is for Nsp1 and Nup98-glyco). Full computational details are available in 'Materials and methods' and *Figure 3—figure supplements 1–5*.

*Figure 3 continued on next page*

*Figure 3 continued*

The following figure supplements are available for figure 3:

**Figure supplement 1.** Scheme illustrating how computational modeling data is presented in the form of maps of the polymer and colloid packing fractions.

**Figure supplement 2.** Computational modeling data for a polymer length equivalent to Nsp1 and colloids of 4.0 nm diameter (equivalent to NTF2 homodimers).

**Figure supplement 3.** Computational modeling data for a polymer length equivalent to Nup98-glyco and colloids of 4.0 nm diameter (equivalent to NTF2 homodimers).

**Figure supplement 4.** Computational modeling data for a polymer length equivalent to reg-FSFG and colloids of 4.0 nm diameter (equivalent to NTF2 homodimers).

**Figure supplement 5.** Computational modeling data for a polymer length equivalent to Nsp1 and colloids of 6.0 nm diameter (equivalent to Impβ).

## Computational modeling – binding of NTRs to FG domain films

Next, we analyzed the binding isotherms. For a given cohesiveness parameter $\varepsilon_{pp}$, we found that the precise setting of the NTR•FG domain interaction $\varepsilon_{pc}$ strongly influenced the amount of bound NTRs for any given NTR concentration, by orders of magnitude for 0.1 $k_BT$ changes in $\varepsilon_{pc}$, in the explored parameter range of 0.1 to 0.5 $k_BT$. It also strongly affected the overall shape of the binding isotherms (*Figure 3—figure supplements 2B*, *3B*, *4B* and *5B*). *Figure 4* (*top row*) shows computational data for parameter sets of $\varepsilon_{pp}$ and $\varepsilon_{pc}$ that best match the experimental results for the different FG domain and NTR types, where $\varepsilon_{pp}$ was determined by the film thickness measurement in the absence of NTRs (see *Figure 3B*). Taking into account that these are fits with a single free parameter ($\varepsilon_{pc}$) over several orders of magnitude in bound NTR and in NTR concentration, the agreement with the experimental data is remarkably good. With $\varepsilon_{pp}$ estimated from the thickness in the absence of NTR (*Figure 3B*) and $\varepsilon_{pc}$ from a comparison to the binding isotherms (*Figure 4*, *top row*), the performance of the model was further validated via the film thickness as a function of NTR concentration in solution. There is good agreement between the experimental data and the computational results (*Figure 4*, *bottom row*). *Table 2* summarizes the results of this analysis, with the estimates of $\varepsilon_{pc}$ varying less than ~20% between the different FG domains and NTRs. Taken together, the measured binding of NTRs to FG domain films was accurately modeled by our simplified description of the relevant interactions in terms of the two key parameters $\varepsilon_{pp}$ and $\varepsilon_{pc}$, where we treat all amino acids in the FG domain chains identically and the NTR surfaces as homogeneous.

It should be emphasized that these interaction strengths represent effective, smeared-out affinities between the polymer beads and colloids in our model (*Figure 3A*). To relate $\varepsilon_{pc}$ to a rough estimate for the binding energy of an NTR in the FG domain film, one may assume the NTR colloid to be surrounded by polymer beads at the maximum polymer packing in our calculations (~20%), yielding at most ~26 and ~53 polymer beads in contact with the colloidal surface, for the NTF2- and Impβ-mimicking colloids, respectively. Hence for NTF2, the corresponding binding energy is $\leq 26 \times \varepsilon_{pc}$; and $\varepsilon_{pc}$ between 0.3 and 0.4 $k_BT$ implies a binding energy of $\leq 10$ $k_BT$ per NTF2 homodimer.

**Table 2.** Interaction parameters determined based on comparison of experiment and computational model.

| FG domain | $\varepsilon_{pp}$ ($k_BT$) | $\varepsilon_{pc}$ ($k_BT$) NTF2 | $\varepsilon_{pc}$ ($k_BT$) Impβ |
|---|---|---|---|
| Nsp1 | 0.024 ± 0.001 | 0.34 ± 0.02 | 0.40 ± 0.02 |
| Nup98-glyco | 0.030 ± 0.002 | 0.36 ± 0.02 | - |
| reg-FSFG | 0.030 ± 0.005 | 0.40 ± 0.02 | - |

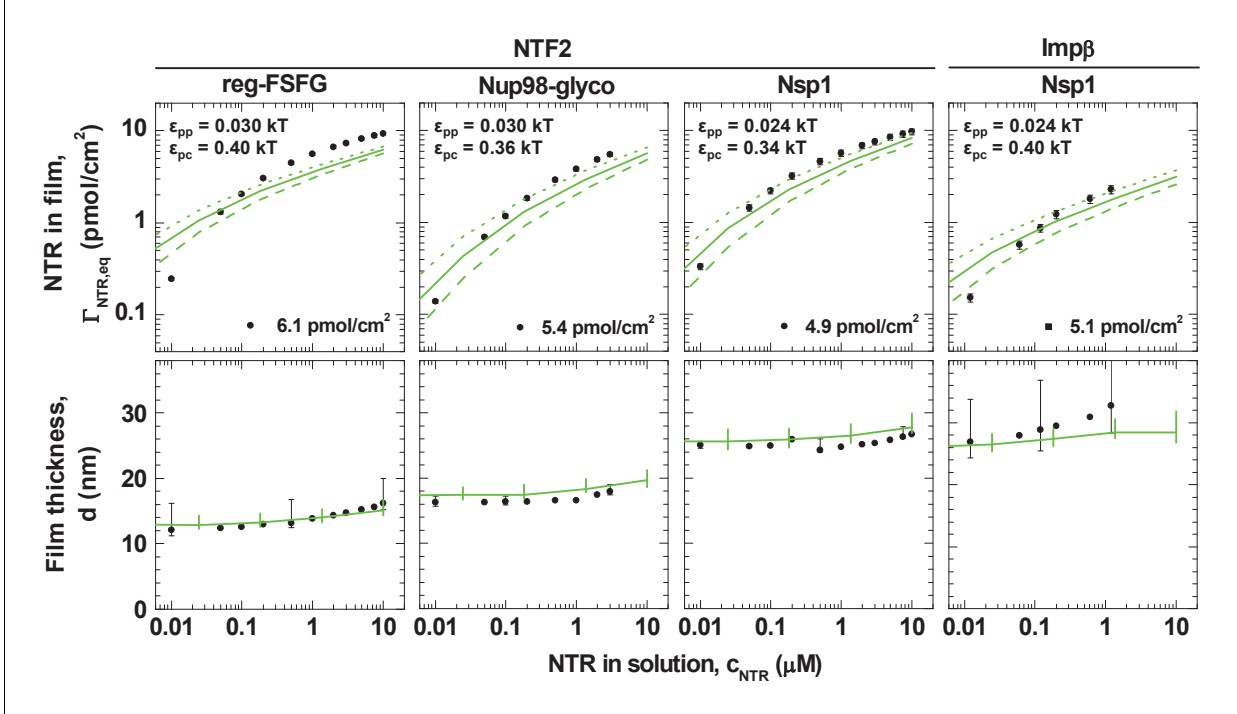

**Figure 4.** Matching of experimental and computational data. The *top row* shows binding isotherms and the *bottom row* the concomitant film thickness evolution. The grafting density was set to 5.5 pmol/cm² in all computations, and the experimental data with the closest FG domain grafting densities are reproduced from *Figure 1* and visualized by *black symbols*. Computational data are shown as *green lines*. The *solid lines* represent the best match to the experiment, and the corresponding $\varepsilon_{pp}$ and $\varepsilon_{pc}$ are indicated. The best match of the binding isotherms was determined by minimization of the least square differences of $\log(\Gamma_{NTR,eq})$ over the range 0.025 μM ≤ $c_{NTR}$ ≤ 10 μM, where the experimental data was interpolated and extrapolated using a linear fit for $c_{NTR} < 0.05$ μM, and the Hill equation for $c_{NTR} > 0.05$ μM, as shown in *Figure 2* and *Figure 2—figure supplement 1*. *Dashed lines* and *dotted lines* in the *top row* correspond to a change in $\varepsilon_{pc}$ by -0.01 $k_B T$ and +0.01 $k_B T$, respectively, with $\varepsilon_{pp}$ unchanged. The lines and upper and lower ends of the vertical bars in the *bottom row* correspond to effective thicknesses where the densities have dropped to 5%, 1% and 10% of the maximal densities in the film, respectively (see *Figure 3* and Materials and methods).

Similarly, $\varepsilon_{pc} \sim 0.4$ $k_B T$ implies a binding energy of ≲20 $k_B T$ per Impβ. Hence our results would correspond to a few $k_B T$ binding energy per FG-binding site on the NTRs, in reasonable agreement with the millimolar affinities per FG motif observed with Impβ (*Milles et al., 2015*).

With the values for $\varepsilon_{pp}$ and $\varepsilon_{pc}$ constrained by the comparison to the experimental data, the computational model makes predictions about the distribution of FG domains and NTRs along the surface normal. These are shown in *Figure 5* (*top row*) for $c_{NTR} = 10$ μM, i.e., in the physiological range of total NTR concentrations. They reveal that, given parameter settings that best match the experimental system (*Table 2*), the NTRs effectively penetrate and fill all FG domain films.

*Figure 5* (*bottom row*) demonstrates that a relatively small change in the FG domain cohesiveness can have a dramatic effect on the NTR distribution. For example, with the NTF2·reg-FSFG interaction maintained at $\varepsilon_{pc} = 0.4$ $k_B T$, an increase in inter-FG domain attraction by 33%, from $\varepsilon_{pp} = 0.03$ $k_B T$ to 0.04 $k_B T$, essentially impaired NTF2 accumulation inside the film, and the NTF2 was enriched instead at the film-solution interface – that is to say, on the film rather than in the film. A similar trend is observed for all other combinations of NTRs and FG domains tested, albeit to a lesser extent for the least cohesive FG domains (Nsp1) with the smaller NTR (NTF2).

The model also indicated that a minor reduction in the colloid·FG domain interaction strength drastically reduces binding (*Figure 3—figure supplements 2*, *3* and *5*). *Figure 6* illustrates this for a selected colloid concentration (1.4 μM, in the range of individual NTR concentrations in the cell) in the solution phase, and shows that by reducing $\varepsilon_{pc}$ by only 25% compared to the best match for the respective NTRs and FG domains, there is a reduction in colloid binding by more than one order of magnitude. Such a dramatic effect is remarkable: For comparison, a reduction of less than 25%

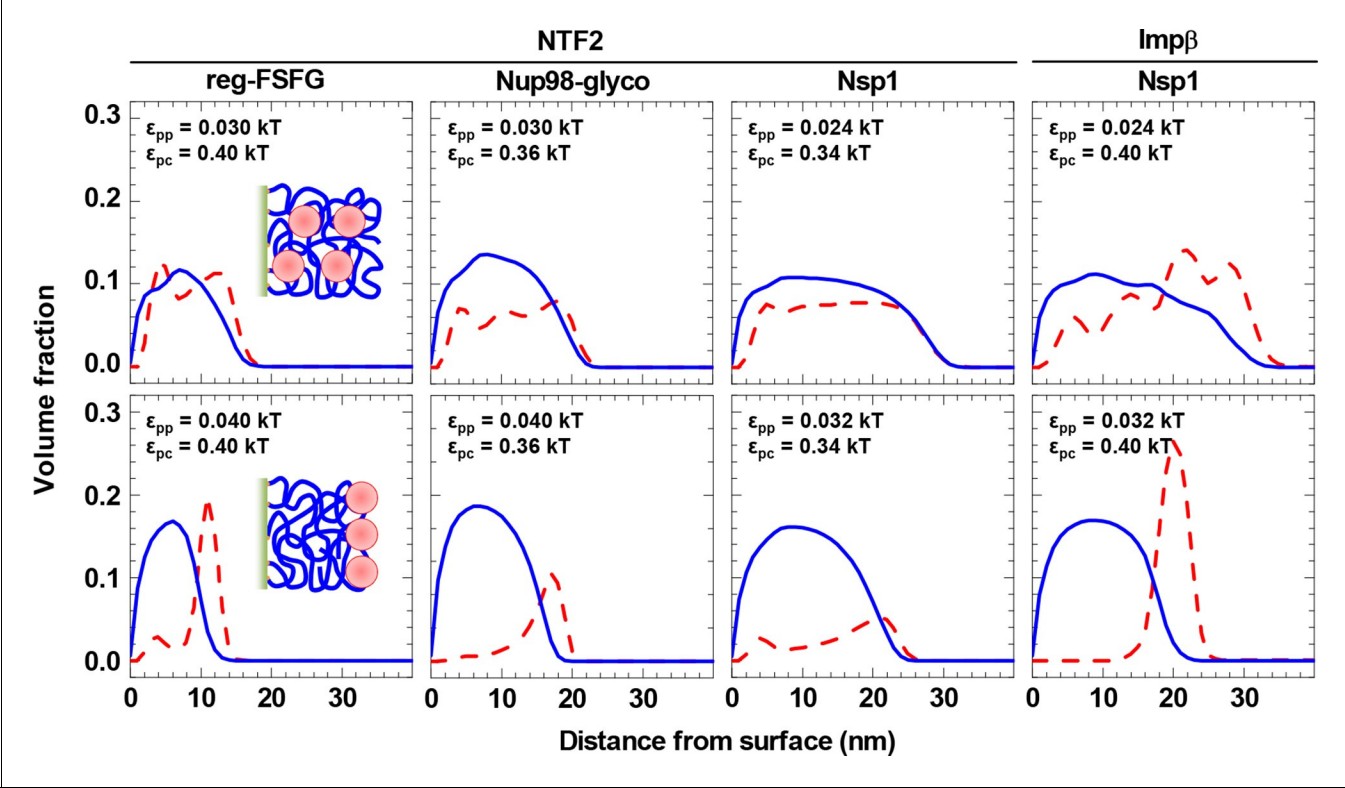

**Figure 5.** NTRs favor the penetration of and binding into FG domain films, but only just so. Computed packing fraction profiles (polymer – *blue solid line*, colloid – *red dashed line*) in the presence of 10 μM NTR, as a function of distance from the grafting surface. The *top row* shows the predictions for the parameter sets of $\varepsilon_{pp}$ and $\varepsilon_{pc}$ that match the experimental data best (cf. *Figure 4* and *Table 2*). The *bottom row* shows predictions with $\varepsilon_{pp}$ increased by 33% compared to the best match. Schemes (*insets*) illustrate the distinct distributions of NTRs with these two parameter choices.

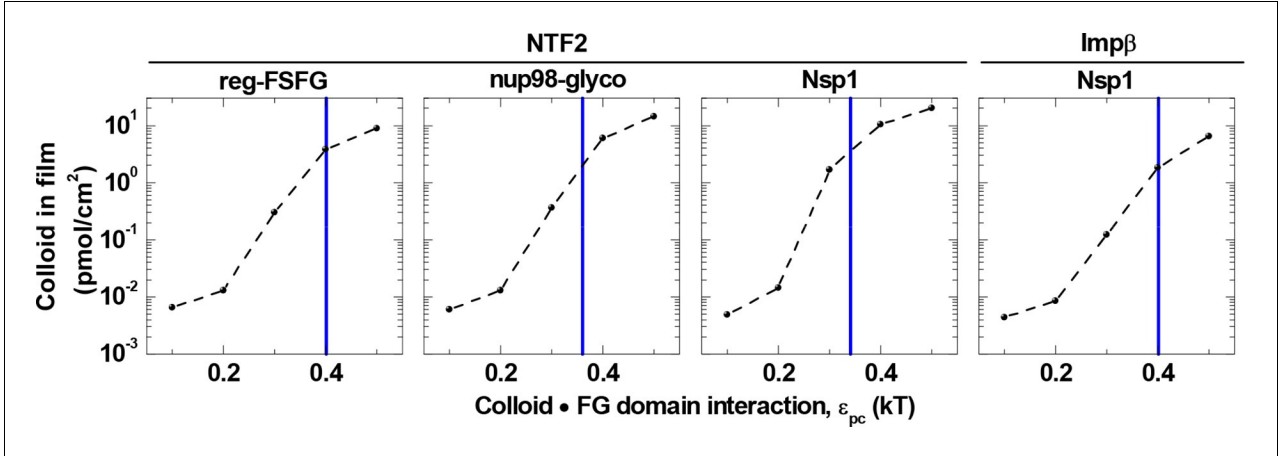

**Figure 6.** Colloid binding depends sharply on colloid•FG domain interaction strength $\varepsilon_{pc}$. Computed colloid binding as a function of $\varepsilon_{pc}$ is shown, for a colloid concentration in solution of 1.4 μM and with the FG domain cohesiveness $\varepsilon_{pp}$ set to the values that best match the experimental data (cf. *Table 2*). The *blue vertical lines* indicate the $\varepsilon_{pc}$ values giving the best match to the experimental data for the indicated FG domains and NTRs (cf. *Table 2*).

would be expected for simple Langmuir-type one-to-one binding. Collectively, these results suggest that the native system is tuned to operate within a rather narrow parameter space in $\varepsilon_{pc}$ and $\varepsilon_{pp}$ that facilitates the strong enrichment of NTRs within the FG domain film, whereas similarly sized proteins with weaker binding strength are effectively excluded.

## Discussion

### FG domain films are significantly compacted compared to non-cohesive polymer brushes

We have measured reconstituted assemblies of FG domains in planar geometry and at grafting densities that are similar to those in the NPC. This yields laterally homogeneous films (*Eisele et al., 2013*), which may be qualified as polymer brushes (*Israelachvili, 1991*). As pointed out previously (*Eisele et al., 2013*), this qualification per se does not imply a distinction between a 'brush-like, entropic' scenario (*Lim et al., 2007*) of the NPC transport barrier on one side, and a hydrogel scenario (*Frey et al., 2006*) on the other. The main distinguishing factor between these opposing scenarios is the level of cohesiveness of the FG domains, which in our computational model is parameterized by $\varepsilon_{pp}$, and defines the compactness of the FG domain phase (*Eisele et al., 2013*). From the comparison between experimental and computational results for the thickness of the different FG domain films (in the absence of NTRs, *Figure 3B*), it appears that the FG domain assemblies are compacted by a factor two to four compared to a film of perfectly non-cohesive ($\varepsilon_{pp} = 0$, equivalent to a self-avoiding random walk model) and flexible peptide chains, as follows from an extrapolation of the computational data (dashed lines in *Figure 3B*). In spite of this condensation, the measured film thickness (*Figure 1*) for Nsp1 is larger than the inner radius of the *S. cerevisiae* NPC (18 to 20 nm [*Alber et al., 2007*; *Yang et al., 1998*]), and that for Nup98-glyco more than 60% of the inner radius of the *X. tropicalis* NPC (~25 nm [*Eibauer et al., 2015*]). This would be consistent with the FG domains forming a pore-filling cohesive meshwork or condensed polymer brush, as implied by the selective phase model (*Frey and Gorlich, 2007*; *Frey et al., 2006*; *Hulsmann et al., 2012*; *Ribbeck and Gorlich, 2001*), at least for Nsp1 in *S. cerevisiae*, and possibly for Nup98-glyco in *X. tropicalis* taking into account the nanopore confinement (*Osmanovic et al., 2012*).

### Universal aspects and negative cooperativity in NTR binding to FG domains

On exposing the FG domain films to NTF2 and Impβ, we find a remarkable quantitative similarity (*Figure 2*) in the binding isotherms for Nsp1, Nup98-glyco, and reg-FSFG, in spite of their chemical diversity. Future studies should test if a different behavior is found for less cohesive FG domains and/or other NTRs. Our analysis in terms of the Hill equation indicates negative cooperativity for NTR binding to the FG domain assemblies. This implies that the free energy of binding per NTR decreases with an increasing amount of bound NTR in the film, and that this decrease is more pronounced than would be expected for the one-to-one binding of NTRs to independent and uncorrelated binding sites (i.e., the Langmuir isotherm). Similar observations have been made previously (*Kapinos et al., 2014*; *Schleicher et al., 2014*; *Wagner et al., 2015*). Key findings of the present work are that this negative cooperativity appears robust against variations in NTR and in the spread of FG motifs between Nsp1, Nup98-glyco, and reg-FSFG, and that it can be reproduced, over orders of magnitude in NTR concentration, by a model that considers the FG domains as homogeneous polymers and the NTRs as featureless spheres, i.e., ignoring chemical and structural heterogeneity.

Our analysis suggests that NTR binding is largely determined by generic effects such as overall binding energy, crowding and excluded-volume interactions, and by the entropic costs of NTR absorption that, in turn, reduce the conformational freedom of the grafted and flexible FG domains. Such effects may well be expected since, at physiological NTR concentrations, the absorption of NTRs roughly doubles the total density of the films (*Figure 5*, *top row*). In our computational model, these effects are included, and the amount of NTR binding follows from collective low-affinity interactions – the balance between the overall cohesiveness of the FG domains ($\varepsilon_{pp}$) and the smeared-out, average NTR•FG domain interactions ($\varepsilon_{pc}$).

## Functional relevance for nucleocytoplasmic transport

In this context, it is important to emphasize that enrichment of NTRs in an FG domain phase – in our model films but also in the NPC – vastly differs from NTR•FG motif binding under dilute conditions. This can be illustrated, for example, by comparing the enrichment of Nsp1 films for NTF2 and Impβ: the higher total Impβ•Nsp1 binding energy (integrated over the Impβ surface) does not translate into a higher enrichment than for NTF2, even in the limit of low NTR concentrations (*Figures 1* and *4*, *top rows*). The binding energy is balanced by excluded-volume interactions and polymer cohesiveness and entropy, i.e., by generic physical effects. At equilibrium, this balance is such that, in spite of the high binding energy between NTRs and FG domains, NTRs can exchange between the NPC and the nucleus/cytoplasm at minimal cost, thus facilitating cargo uptake and release.

Remarkably, this overall balance appears rather finely tuned in several respects. Firstly, a small reduction in $\varepsilon_{pc}$ produces a dramatic decrease in the uptake of colloids by the FG domain films (*Figure 6*). This is of major functional importance: by its strong dependence on $\varepsilon_{pc}$, the variable colloid uptake explains how FG domain assemblies in the NPC greatly favor the uptake of NTRs over other cytosolic proteins, which have been inferred to also bind FG domains (*Hough et al., 2015*), albeit more weakly. This then explains how, because of the selective uptake, NTRs can efficiently translocate across the NPC while more weakly binding proteins cannot. We anticipate that this predicted fine tuning can be experimentally validated by systematically adjusting protein affinity to FG domain assemblies, which awaits further studies.

In addition, from our comparison between experiment and computational modeling, the NTRs appear to favor the penetration of and binding into the FG domain assemblies, but only just so: penetration is inhibited for slightly larger FG domain cohesiveness (i.e., larger $\varepsilon_{pp}$), which results in NTRs preferentially binding on top of (and not into) the FG domain assembly (*Figure 5*). This is consistent with the remarkable selectivity exhibited by NPC transport and suggests that the cohesiveness of the FG domains is tuned to be as tight as possible to optimize exclusion of inert proteins by their size (*Eisele et al., 2013*), while still sufficiently loose to facilitate penetration by NTRs. These features also lend themselves to further experimental validation: the model predicts how NTRs are distributed in FG domain assemblies and how these distributions depend on the interaction strength between NTRs and FG domains, which can be verified in future neutron or X-ray reflectometry measurements on FG domain films.

Extrapolating our results on planar assemblies to the pore geometry of the NPC, we note that large structural changes can occur within the here determined parameter range of $\varepsilon_{pp}$: In analogous calculations for FG domains in an NPC-mimicking pore geometry of 50 nm diameter (*Osmanovic et al., 2012*; *2013b*), we observed a transition (*Osmanovic et al., 2012*) between, on one hand, a central and pore-occluding condensate of FG domains in the NPC conduit, and on the other hand, a more open state with FG domains localized closer to the pore wall (see, e.g., Figure 7 in *Osmanovic et al., 2013b*). Such large and collective transitions are required to facilitate transport of larger cargo•NTR complexes, with a size comparable to the nuclear pore diameter, through the NPC.

## Towards a minimal physical model for the NPC

Depending on the interaction strengths, FG domains assemblies may thus adopt qualitatively different behaviors, e.g., *ab*sorption versus *ad*sorption of NTRs and different types of polymer condensation. The overall FG domain interactions and NTR•FG affinity appear to be tuned close to the boundaries that separate these types of behaviors. The observed sensitivity to the values assigned to parameters offers an explanation of why different modeling approaches thus far have led to qualitatively different predictions of FG domain behavior in the NPC (*Ando et al., 2013*; *2014*; *Gamini et al., 2014*; *Ghavami et al., 2014*; *Miao and Schulten, 2009*; *Mincer and Simon, 2011*; *Moussavi-Baygi et al., 2011a*; *2011b*; *Opferman et al., 2012*; *2013*; *Osmanovic et al., 2012*; *2013b*; *Popken et al., 2015*; *Tagliazucchi et al., 2013*; *Wolf and Mofrad, 2008*), where most models succeed in capturing at least some aspects of experimental data on NPCs. The here observed sensitivity to parameter settings indicates that it is critical to calibrate computational models and their parameters against well-controlled experiments. We propose that experimental data obtained on well-defined FG domain assemblies (such as those provided and used here), possibly complemented by structural data of isolated FG domains in solution (*Yamada et al., 2010*), could

serve as reference for such calibration. It would be desirable that identical sets of reference data are used by the computational modelers, as this would enable rigorous comparison between different computational approaches. To facilitate this effort, we provide data files of the NTR binding and thickness isotherms (*Figure 1—source data 1*).

Our results have the advantage that they allow for a quantitative comparison between computational simulations and the experimental (model) system. For the entire NPC, such rigorous testing of computational models is presently complicated by experimental uncertainties in the locations of different FG domains inside the NPC, by the difficulties in accurately validating interaction parameters for the ensemble of FG domains in the NPC, and by the predicted bistable behavior of polymers grafted in nanopore geometries (see, e.g., *Peleg et al., 2011* and *Osmanovic et al., 2012*). That said, given the observed insensitivity to chemical heterogeneity of the FG domains, one can estimate the level of detail that will need to be included for building an appropriate model and understanding of mechanisms of selective transport in the NPC. The results presented here indicate that it is essential to take into account the flexible nature and cohesion of FG domains, as well as the crowding of NTRs that bind to the FG domain assemblies, but that heterogeneity at the scale of amino acids may only be of minor importance.

## Conclusion

In summary, we have used a bottom-up nanoscale system for a quantitative study of how NTRs interact with FG domains from the NPC. Highly similar binding isotherms were found for NTF2 binding to assemblies of Nsp1 from *S. cerevisiae*, of Nup98-glyco from *X. tropicalis*, and of an artificially designed regular FSFG construct; and for a different NTR, Impβ, binding to Nsp1. This similarity suggests that – while the overall balance of interactions is essential – the detailed chemical and structural heterogeneity of the FG domains is not a critical factor for how NTRs interact with FG domain assemblies and thus with the NPC. This conclusion is supported by the good agreement – over several orders of magnitude of NTR concentration – between the experimental data and a physical model that treats the FG domains as chemically and structurally homogeneous polymers and the NTRs as spherical colloids.

These results imply that the enrichment of NTRs into the FG domain phase is determined by generic physical effects – the flexible nature and spatial confinement of FG domains and the NTR size – and by the overall balance of a collection of low-affinity inter-FG-domain and NTR•FG domain interactions. Moreover, our computational data show that moderate changes in this overall balance cause remarkably large changes in the protein uptake by the FG domain assemblies, an observation that is fully consistent with the transport selectivity of the NPC.

Given the success of our model in replicating NTR binding behavior in a nanoscale mimic for the NPC, we therefore propose that a similar approach may be viable to describe NPC transport selectivity, i.e., in terms of generic polymer models, without necessarily taking into account the full amino acid sequences of FG domains and NTRs. However, our results also show that computational models need to be carefully calibrated to experimental data – such as has been done in this work – if they are to provide a meaningful contribution to the NPC field, since small ($\sim 30\%$) changes in interaction parameters can result in qualitatively different behaviors.

## Materials and methods

### Proteins and buffers

We used the following FG domains: Nsp1 (64.1 kDa), amino acids 2 to 601 of Nsp1 with a C-terminal His$_{10}$ tag; Nup98-glyco (58.3 kDa), amino acids 1 to 485 of Nup98 with $\sim 30$ O-GlcNAc modified S and T residues per chain (*Eisele et al., 2013*; *Labokha et al., 2013*) and an N-terminal His$_{14}$-TEV tag; reg-FSFG (34.1 kDa), an artificially designed regular FSFG domain with 16 repetitions of the sequence STPA**FSFG**ASNNNSTNNGT and an N-terminal His$_{14}$-TEV tag; reg-SSSG (32.2 kDa), a polypeptide identical to reg-FSFG but with phenylalanines replaced by serines. All FG domains were purified as described earlier (*Eisele et al., 2010*; *2013*; *Frey et al., 2006*; *Labokha et al., 2013*) and stored at a concentration of 10 mg/mL in 50 mM Tris pH 8.0, 6 M guanidine hydrochloride (GuHCl) at –80°C. We used the following NTRs: NTF2 from *H. sapiens* (NTF2, amino acids 1 to 127; 29.0 kDa for the homodimer); and Impβ from *S. cerevisiae* (95.2 kDa). NTF2, the W7A mutant of NTF2 and

Impβ were expressed and purified as previously described (*Bayliss et al., 1999*; *Eisele et al., 2010*) and stored at a concentration of 100 µM in working buffer (10 mM Hepes, pH 7.4, 150 mM NaCl) at −80°C. Before use, all protein constructs were diluted in working buffer to desired concentrations. For all our measurements, the residual concentration of GuHCl in the final solution was below 60 mM. The purity of proteins was verified by SDS-PAGE (*Figure 1—figure supplement 1*).

## In situ combination of spectroscopic ellipsometry (SE) and quartz crystal microbalance with dissipation monitoring (QCM-D)

The formation of FG domain films and the binding of NTRs to FG domain films were simultaneously followed by SE and QCM-D on the same surface and in a liquid environment (*Figure 1—figure supplement 3A*) (*Richter et al., 2013*). To this end, we used a custom-built cuvette-like open fluid cell, placed in a Q-Sense E1 system (Biolin Scientific AB, Västra Frölunda, Sweden; providing QCM-D data) and mounted on a spectroscopic rotating-compensator ellipsometer (M2000V, J. A. Woollam Co., Lincoln, NE; providing SE data), as described in detail elsewhere (*Carton et al., 2010*).

In the SE measurements, ellipsometric angles ($\Delta$ and $\psi$) were acquired over a wavelength range of $\lambda$ = 380 to 1000 nm at 70° angle of incidence and about 5 s time resolution. In the QCM-D measurements, frequency and dissipation shifts ($\Delta f_i$ and $\Delta D_i$) were acquired for six overtones ($i$ = 3, 5,... , 13; corresponding to resonance frequencies of $f_i \approx$ 15, 25,... , 65 MHz) with a time resolution better than 1 s. Prior to each measurement, the walls of the cuvette were passivated by incubation with a buffer solution containing 10 mg/mL of bovine serum albumin (BSA; Sigma) for 30 min. The cuvette was rinsed with buffer, ultrapure water and blow-dried with nitrogen. For the measurement, the cuvette was filled with ∼2 mL working buffer, continuously stirred and held at a temperature of 23°C. Samples were injected directly into the buffer-filled cuvette at desired concentrations. To remove samples, the cuvette content was diluted by repeated addition of excess buffer and removal of excess liquid until the concentration of soluble sample, estimated from the dilution rate, was below 10 ng/mL.

## Surface functionalization, FG domain film formation, and titration with NTF2

For measurements with NTF2 and Nsp1, we used His tag capturing QCM-D sensors (QSX340; Biolin Scientific AB). These sensors are coated with a thin layer of poly(ethylene glycol) (PEG) that exposes $Cu^{2+}$ ions for the capture of His tagged molecules, and could be readily used as provided for FG domain film formation. We previously demonstrated that His tag capturing QCM-D sensors are suited to create dense monolayers of site-specifically anchored Nsp1, and that such Nsp1 films have comparable properties to Nsp1 films formed on functionalized supported lipid bilayers (SLBs) (*Eisele et al., 2012*). For measurements with Nup98-glyco, with reg-FSFG, and with Impβ and Nsp1, we used SLBs as immobilization platform instead as this provided improved binding specificity. These measurements were performed on silica-coated QCM-D sensors that are optimized for combined QCM-D/SE experiments (QSX335; Biolin Scientific AB). The sensors were cleaned by immersion in a 2% sodium dodecyl sulfate solution for 30 min, rinsed with ultrapure water, blow-dried with nitrogen, and exposed to UV/ozone (BioForce Nanosciences, Ames, IA) for 30 min. We mounted the cleaned sensors in the combined SE/QCM-D and functionalized their surface with supported lipid bilayers (SLBs) exposing $Ni^{2+}$ ions for the capture of His tagged molecules, as described previously (*Eisele et al., 2010*; *2013*). Briefly, we used sonication to prepare small unilamellar lipid vesicles (SUVs) containing dioleoylphosphatidylcholine (DOPC; Avanti Polar Lipids, Alabaster, AL) and 3 to 10 mol-% of lipid analogs with headgroups comprising two or three $Ni^{2+}$-chelating nitrilotriacetic acid moieties (bis-NTA or tris-NTA) (*Beutel et al., 2014*; *Lata et al., 2006*). SLBs were spontaneously formed by injecting SUVs (at 50 µg/mL final concentration) with $NiCl_2$ (at 10 µM final concentration) into the buffer-filled SE/QCM-D fluid cell. SLB formation was monitored by SE and QCM-D and only SLBs of good quality (i.e., showing low QCM-D dissipation shifts, $\Delta D < 0.5 \times 10^{-6}$, and high frequency shifts, $|\Delta f| > 25$ Hz) were used for further measurements.

We formed FG domain films by injecting the FG domains directly into the SE/QCM-D cuvette equipped with a functionalized sensor. FG domain film formation was monitored, and FG domain concentration (up to 2.9 µM) and incubation time (up to 90 min) modulated to obtain FG domain films of desired grafting density (*Figure 1—figure supplement 3B*). NTRs were titrated in discrete

steps, first increasing and then decreasing, followed by at least 5 min of continuous rinsing with working buffer to remove NTRs from the solution phase. Incubation times were 5 to 30 min for titration steps with increasing NTR concentrations and 5 min for decreasing concentrations, i. e., sufficiently long for equilibrium to be reached, as verified from the SE/QCM-D curves (*Figure 1— figure supplement 3C*).

## Quantification of film thickness

We determined the thickness of FG domain films by fitting the QCM-D data to a continuum viscoelastic model, as described in detail previously (*Eisele et al., 2012*). Briefly, we used the software QTM (Johannsmann) (option "small load approximation" (*Johannsmann, 1999*; *2008*)). The FG domain films were modeled as homogeneous viscoelastic films with a storage modulus ($G'$) and a loss modulus ($G''$) that depend on frequency in the form of a power law. The film density was fixed based on the areal mass density (determined by SE, see below) and partial specific volume of proteins, and the density of water. The semi-infinite bulk solution was assumed to be a Newtonian fluid with the density and viscosity of water.

The interface between the FG domain film and the bulk solution is not ideally sharp, and the definition of film thickness thus not trivial. The viscoelastic model neglects the fuzzy interface and assumes a homogeneous film. However, because the acoustic contrast in polymer materials is generally high (*Johannsmann, 2008*), the (acoustic) thickness measured by QCM-D is expected to include a substantial part of the interfacial region that has a relatively low polymer density (*Domack et al., 1997*). The specified errors represent a confidence level of one standard deviation (68%). In previous studies (*Eisele et al., 2010*; *2012*; *2013*), we found that the thickness results obtained by QCM-D for FG domain films are comparable to within the specified confidence levels to those obtained with other techniques (atomic force microscopy and SE), thus confirming that the thickness determination is robust.

## Quantification of adsorbed amounts

SE data were fitted to a model of multiple optically homogeneous layers, implemented in the software CompleteEASE (J. A. Woollam Co.), to quantify protein surface densities. The fitting methods for QSX335 and QSX340 sensor substrates are described in detail in refs. (*Carton et al., 2010*) and (*Eisele et al., 2012*), respectively. Irrespective of the substrate, the FG domain film was treated as a transparent Cauchy film with an effective optical thickness $d_{SE}$ and a wavelength-dependent refractive index $n(\lambda)$. Protein surface densities $\Gamma$ where obtained through de Fejter's equation (*De Feijter et al., 1978*), i.e., $\Gamma = d_{SE}\Delta n / (M_W \times dn/dc)$, where $\Delta n$ is the difference in refractive index between the FG domain film and the buffer solution (assumed to be wavelength independent) and $M_W$ the protein molecular mass. We used $dn/dc = 0.18$ cm$^3$/g as refractive index increment for our protein films (*Richter et al., 2013*). The resolution in $\Gamma \times M_W$ was typically 0.5 ng/cm$^2$. Among the optical mass-sensitive techniques, SE is particularly suited to quantify the areal mass density of organic films up to a few 10 nm thick, because mass determination is virtually insensitive to the distribution of material within the film (*Richter et al., 2013*).

## Computational model

To model the experimental systems, we adapted a classical density functional theory approach that was derived, validated against Monte Carlo simulations, and applied in previous studies of polymers in a cylindrical confinement (*Osmanovic et al., 2012*; *2013b*). We described the FG domain assembly as a collection of one-end-grafted polymers anchored homogeneously to the bottom of a cylinder of 100 nm in diameter and 120 nm in height (*Figure 3A*). Each polymer was modeled as a string of physically and chemically identical beads that can rotate freely with respect to each other. The bead diameter and (fixed) bond length of the polymers were both taken to be 0.76 nm, such that each bead is in direct contact with its nearest neighbors on the string. Specifically, this yields a Kuhn length of 0.76 nm, i.e., twice the contour length of an amino acid, and a persistence length of 0.38 nm, mirroring the flexibility of unfolded polypeptide chains (*Stirnemann et al., 2013*). The number of beads per polymer chain was chosen to be 300, 260 and 155 to represent Nsp1, Nup98-glyco and reg-FSFG, respectively. These numbers match half the number of amino acids (except the His tag used for anchorage) of the respective FG domain constructs being modeled, and thus the

contour length, to within 5% (*Table 1*). The polymer grafting density was set to 5.5 pmol/cm$^2$, i.e., 3.3 polymers per 100 nm$^2$, to approximately match the FG domain grafting density in the NPC and the majority of the experimental data sets presented here. It was kept identical for the different FG domains to facilitate comparison and cross-validation of the computational results. For significantly lower grafting densities, lateral heterogeneity became increasingly important – as also observed experimentally (*Eisele et al., 2013*) – partly invalidating our approach in that regime. On the other hand, for significantly higher grafting densities, excluded-volume interactions were such that the computational convergence was much harder to achieve. NTF2 was represented as a colloidal sphere of 4.0 nm diameter, where the latter value was based on the diameter of a sphere that approximately includes the atomistic NTF2 dimer structure (*Bayliss et al., 1999*). Similarly, Impβ was represented as a sphere of 6.0 nm, roughly matching the atomistic Impβ structure (*Forwood et al., 2010*), and 50% larger than NTF2, in agreement with a difference in diameter expected based on the mass difference between Impβ and NTF2. The colloidal spheres could freely diffuse into and out of the cylinder with a chemical potential corresponding to the molar concentration, as described previously (*Osmanovic et al., 2013b*).

The affinity between two polymer beads was parameterized by $\varepsilon_{pp}$, and the interaction between a polymer bead and a colloid by $\varepsilon_{pc}$, where $\varepsilon_{pp}$ and $\varepsilon_{pc}$ refer to the energy gain on bringing two polymer beads, and a polymer bead and a colloid, respectively, from infinite separation to hard contact. The attractive interactions were of the generic, exponential form with a decay length of 1 nm. $\varepsilon_{pp}$ and $\varepsilon_{pc}$ were varied to map the morphological space of the FG domain/NTR films as a function of the balance between inter and intra FG domain interactions on the one hand, and FG domain-NTR interactions on the other. We assumed that there was no attraction between the colloids.

Rotational symmetry was assumed around the central axis of the cylinder (perpendicular to the grafting surface), thus reducing the system to two dimensions for computational efficiency. For consistency with our previously developed algorithms, zero-density boundary conditions were imposed for both polymers and colloids at the side walls of the cylinder. At the cylinder top and bottom, zero-density boundary conditions were used for the polymers and periodic boundary conditions for the colloids.

Local (and a priori radially dependent) film thicknesses were defined via iso-density profiles corresponding to a defined fraction (1%, 5% and 10% ) of the maximal density within the film. For comparison to the experimental data, these thicknesses were averaged across the cylinder (taking into account the $2\pi r$ weighting factor in cylindrical coordinates). The results were verified for robustness against the exact threshold setting (1%, 5% and 10%, see Results section), and found to typically correspond to the thickness that includes ≥90% of the total film material.

Amounts of bound colloids were expressed as areal densities averaged over the full surface (of 100 nm diameter) of the cylinder. To test for boundary-related artifacts in the computational results, areal colloid densities were also computed as averages across the central part (of 50 nm diameter) of the cylinder. Within the range of parameters relevant for comparison with experiment, the results agreed to within less than a factor of two.

Polymer and colloid density profiles were determined from respective packing fractions versus the axis that is normal to the grafting surface. Analysis of a profile was performed on the average of profiles for different radial positions in the cylinder.

## Acknowledgements

We thank Ariberto Fassati and Ian J Ford (both UCL) for fruitful discussions, Totta Kasemo (CIC biomaGUNE) for measurements and data analysis, and Neil Marshall and Anna Stamp (both MRC-LMB) for producing NTF2 protein and the NTF2 W7A mutant.

## Additional information

### Funding

| Funder | Grant reference number | Author |
| --- | --- | --- |
| European Commission | FP7-PEOPLE-2012-IEF-327975 | Raphael Zahn |

| European Research Council | FP7-ERC-2012-StG-306435 | Ralf P Richter |
| Dr. Mortimer and Theresa Sackler Foundation | | Dino Osmanović Bart W Hoogenboom |
| Deutsche Forschungsgemeinschaft | SFB 944 | Changjiang You |
| Medical Research Council | Grant U105178939 | Murray Stewart |
| Spanish Ministry for Economy and Competitiveness | MAT2011-24306 | Ralf P Richter |

The funders had no role in study design, data collection and interpretation, or the decision to submit the work for publication.

#### Author contributions

RZ, Conceived and designed research, Acquired data, Analyzed data, Wrote the paper; DO, Conceived and designed research, Acquired data, Analyzed data; SE, CAC, Acquired data, Analyzed data; SF, MS, CY, DG, Contributed new reagents/analytic tools; BWH, RPR, Conceived and designed research, Analyzed data, Wrote the paper

#### Author ORCIDs

Raphael Zahn, http://orcid.org/0000-0003-0223-0697
Bart W Hoogenboom, http://orcid.org/0000-0002-8882-4324
Ralf P Richter, http://orcid.org/0000-0003-3071-2837

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
