## [Decision Letter]

Thank you for submitting your article "A minimal physical model describing the interaction of nuclear transport receptors with FG nucleoporin domain assemblies" for consideration by *eLife*. Your article has been reviewed by two peer reviewers, and the evaluation has been overseen by a Reviewing Editor and Vivek Malhotra as the Senior Editor.

The reviewers have discussed the reviews with one another and the Reviewing Editor has drafted this decision to help you prepare a revised submission.

We would caution using the current title, since the theory on which DFT is based is far from trivial, and just because they can model based on this "complex model" with very few parameters a few experimental observables (binding isotherms etc.), I would not be certain that this is the "minimalistic" model. Other computational models have also been able to capture certain experimental results (e.g. Popken et al., 2015).

Summary:

The work is of high technical quality and it is interesting that the computational model, which in their previous work was used to describe AFM data on physiological NPCs, can capture experiments with different NTR-FG systems. Overall, the data in Figure 1 and the supplementary figures associated with Figure 1 are very interesting. They appear to provide definitive evidence against large-scale conformational changes that have been proposed to accompany NTR interactions with FG domains. Additionally, these data, which have been modeled in Figure 2, appear to yield isotherms of similar shapes for different combinations of NTR and FG domains. The reviewers agree that your work is potentially of interest for *eLife*. However, there are several issues that remain unanswered that will have to be addressed to make the paper suitable for publication. The essential revisions required follow.

Essential revisions:

1) The Hill analysis yields Hill coefficients that are less than unity and this is thought to imply negative cooperativity. It is worth emphasizing that a Hill coefficient that is less than unity is necessary but not sufficient to invoke negative cooperativity. Further, the modeling does not seem to provide a rationalization for the proposed negative cooperativity.

2) As for the model itself, it is unclear if the beads are intended to represent FG motifs or collections of residues that incorporate multiple FG motifs. This is relevant because the question is if the energy scale for polymer-polymer interactions includes only FG motifs. If yes, then what is the basis for ignoring interactions that do not involve FG motifs?

3) The choice of the polymer model needs further rationalization. In particular, is it possible to obtain equivalent fits with a polymer model that belongs to a completely different universality class such as a self-avoiding random walk for example? A suitable set of reference models needs to be constructed, as should a suitable random prior to understand why one should put faith in the model.

4) The manuscript ends rather abruptly as far as pushing the model is concerned. Although the matching of experimental data is nice, the predictive insights that emerge from the model remain unclear. This is important because one would like to know specific predictions in order to design ways to test the merits of the model.

5) Muthukumar has developed an elegant family of theories for macromolecular translocation and affords several analytical routes to make specific predictions through analysis of the simulation results.

---

## [Author Response]

*We would caution using the current title, since the theory on which DFT is based is far from trivial, and just because they can model based on this "complex model" with very few parameters a few experimental observables (binding isotherms etc.), I would not be certain that this is the "minimalistic" model. Other computational models have also been able to capture certain experimental results (e.g. Popken et al., 2015).* Our model is minimal in the way it approaches nucleoporins as homogeneous polymers and transport receptors as spherical colloids. We consider this minimal in the scientific context relevant for the NPC. However, the notion of “minimal” or “minimalistic” clearly depends on the reader’s perspective. We have therefore followed the reviewers’ recommendation and have removed “minimal” from the title, as well as at several occurrences in the manuscript.

Moreover, we feel that our model does considerably more than describing “a few experimental observables”. The model describes quantitatively the overall binding isotherms for different nucleoporins and nuclear transport receptors, together with also accounting for the overall resulting change in morphology (i.e., thickness) of precisely defined and characterised FG domain assemblies. Moreover, it does so over several orders of magnitude of receptor concentration. We believe that this represents a considerable advance towards understanding the transport selectivity of the nuclear pore complex (NPC).

In addition, we have made more explicit that other computational models (incl. Popken et al., 2015) have also been able to capture certain experimental results, in the section “Towards a minimal physical model for the NPC”.

Essential revisions:

*1) The Hill analysis yields Hill coefficients that are less than unity and this is thought to imply negative cooperativity. It is worth emphasizing that a Hill coefficient that is less than unity is necessary but not sufficient to invoke negative cooperativity. Further, the modeling does not seem to provide a rationalization for the proposed negative cooperativity.* We thank the reviewers for this question regarding the negative cooperativity and its implications for the interactions involved in transport through NPCs. We followed the standard textbook definitions when describing the binding as negatively cooperative when the Hill coefficients are less than unity. For example, Voet D & Voet JG, Biochemistry, 4^th^ ed. (John Wiley & Sons, 2011), specifies: “The quantity *n*, the Hill constant, increases with the degree of cooperativity of a reaction and thereby provides a convenient, although simplistic, characterization of a ligand-binding reaction. […] If *n* < 1, the reaction is termed negatively cooperative”. Because of the rather simplistic nature of this definition, in the text we explicitly describe the meaning of “negatively cooperative” in the context of our work and say “that the free energy of binding per NTR decreases with an increasing amount of bound NTR in the film, and that this decrease is more pronounced than would be expected for the one-to-one binding of NTRs to independent and uncorrelated binding sites (i.e., the Langmuir isotherm)”. We included this more precise definition in the section “Universal aspects and negative cooperativity in NTR binding to FG domains”.

We have also clarified the rationalisation of the negative cooperativity in the same section (“Universal aspects and negative cooperativity in NTR binding to FG domains”), where we discuss the generic nature of the negative cooperativity and attribute it to generic polymer behaviour of the FG domains. Specifically, we infer that, beside the binding energy (that is included in simple Langmuir behaviour), NTR binding is affected by “crowding and excluded-volume interactions, and by the entropic costs of NTR absorption that, in turn, reduce the conformational freedom of the grafted and flexible FG domains”.

*2) As for the model itself, it is unclear if the beads are intended to represent FG motifs or collections of residues that incorporate multiple FG motifs. This is relevant because the question is if the energy scale for polymer-polymer interactions includes only FG motifs. If yes, then what is the basis for ignoring interactions that do not involve FG motifs?* We are grateful for this suggestion and have now expanded the text in the section “Computational approach and cohesiveness of the FG domain films” to clarify the nature of the beads.

Specifically, the beads are not intended to represent specific FG motifs nor necessarily collections of residues that incorporate multiple FG motifs.

The model considers the polymers as homogeneous, ignoring the exact distribution of FG motifs in the FG domains. Polymer-polymer interactions are thus considered smeared out along the polymer. They are included in one effective, overall attractive phenomenological interaction parameter ε_pp_ for the pair-potential between two beads, in addition to the excluded-volume interactions, which are treated separately (as detailed in Osmanovic et al., 2013b). The parameter ε_pp_ thus effectively includes interactions between FG motifs but also interactions with other parts of the FG domain.

*3) The choice of the polymer model needs further rationalization. In particular, is it possible to obtain equivalent fits with a polymer model that belongs to a completely different universality class such as a self-avoiding random walk for example? A suitable set of reference models needs to be constructed, as should a suitable random prior to understand why one should put faith in the model.* The choice of the polymer model used in this manuscript is based on our earlier work ([59]; [61], including the supplementary material) in which the polymer model and polymer-colloid model were extensively tested and validated against Monte Carlo simulations. We have expanded the text (in the section “Computational model”) to discuss this point more extensively. In response to the reviewers’ comment about different universality classes, we would like to emphasise that our model encompasses different types of polymer behaviour and indeed our manuscript does not make any claim about FG domain behaviour belonging to one particular universality class or another. These types of behaviour are generally accounted for by our model, depending on the particular parameter settings employed. For example, the self-avoiding random walk model raised by the reviewers corresponds to our polymer model with ε_pp_ = 0 (just hard-sphere interactions). However, this model does not provide a good match to the experimental data as can be inferred from the revised section “FG domain films are significantly compacted compared to non-cohesive polymer brushes”.

*4) The manuscript ends rather abruptly as far as pushing the model is concerned. Although the matching of experimental data is nice, the predictive insights that emerge from the model remain unclear. This is important because one would like to know specific predictions in order to design ways to test the merits of the model.* We are grateful for this suggestion and have expanded the text to outline more fully the implications of the modelling for nucleocytoplasmic transport. In our view, one of the most exciting perspectives is to predict and to test transport selectivity through the NPC, along the lines we discuss in the final paragraph of the section “Towards a minimal physical model for the NPC”. We have expanded our original discussion here and make more explicit our prediction that this transport selectivity may be modelled and understood by taking into account the flexible nature and cohesion of FG domains and crowding alone, so that one can circumvent the complications inherent in including precise details of much of the underlying chemistry and sub-molecular structure.

Currently a rigorous testing of this prediction is complicated, until more data becomes available, by experimental uncertainties in, for example, the locations of different FG domains inside the NPC, the difficulties in accurately validating interaction parameters for the ensemble of FG domains in the NPC, together with the predicted bistable behaviour of polymers grafted in nanopore geometries (see, e.g., [63] and [59]). That said, our model makes a number of testable predictions that should provide a spur for further studies. Consequently, we have revised and expanded the Discussion to describe more fully how the predictions of the model can be tested with additional experiments, in particular the distributions of NTRs in FG domain films (Figure 5) and the extreme dependence on protein uptake on the protein affinity for the FG domains (Figure 6). This has now been made more explicit in the section “Functional relevance for nucleocytoplasmic transport”.

Finally, we note that our manuscript is largely about testing the merits of the model in predicting how nuclear transport receptors bind to FG domain assemblies, and that our model passes that test remarkably well.

5) Muthukumar has developed an elegant family of theories for macromolecular translocation and affords several analytical routes to make specific predictions through analysis of the simulation results.

Muthukumar’s pioneering work (see, for example, his review in Protein Pept Lett. 2014. 21:209-216) refers to the uptake and translocation of flexible polymers (i.e., unfolded macromolecules) in solid-state and α-haemolysin nanopores and, although it represents a crucial advance in this area, it refers to a quite different type of transport to that seen with NPCs. Thus, transport through NPCs generally concerns the translocation of folded, globular proteins, and so does the uptake of transport receptors in our FG domain assemblies and consequently is on a different scale to the passage of flexible polymers through very narrow pores. Consequently we feel that Muthukumar’s work is not strictly relevant to the system that is being investigated in our manuscript, but would be delighted to follow more specific guidance by the reviewers on this topic.